# Risk Assessment and Determination of Heavy Metals in Home Meal Replacement Products by Using Inductively Coupled Plasma Mass Spectrometry and Direct Mercury Analyzer

**DOI:** 10.3390/foods11040504

**Published:** 2022-02-10

**Authors:** Hee-Jeong Hwang, Gyo-Ha Hwang, So-Min Ahn, Yong-Yeon Kim, Han-Seung Shin

**Affiliations:** 1Research Institute of Integrative Life Sciences, Dongguk University-Seoul, 32, Dongguk-ro, Ilsandong-gu, Goyang-si 10326, Gyeonggi-do, Korea; piatop@hanmail.net; 2Department of Food Science and Biotechnology, Dongguk University-Seoul, 32, Dongguk-ro, Ilsandong-gu, Goyang-si 10326, Gyeonggi-do, Korea; hkh1993@naver.com (G.-H.H.); syjakeun@naver.com (S.-M.A.); kimyy613@naver.com (Y.-Y.K.)

**Keywords:** home meal replacement, heavy metal, inductively coupled plasma mass spectrometry, direct mercury analyzer, risk assessment

## Abstract

This study quantified six heavy metals (Pb, Cd, As, Sn, Hg, and Me-Hg) in home meal replacement products. Satisfactory linearity (*R*^2^ > 0.99), recovery (80.65–118.02%), limits of detection (0.02–2.81 μg/kg), limits of quantification (0.05–8.51 μg/kg), accuracy (80.49–119.87%), precision (0.26–14.93%), standard uncertainty (0.082–0.321%) and relative standard uncertainty (0.084–0.320%) of the six heavy metals were obtained. The average concentration of the six heavy metals was 8.87 μg/kg. Heavy metal concentrations were converted to food intake data of 0.009 μg/kg to recalculate the 95th percentile food intake data (g/day) of individual heavy metals. These were then divided by age group to evaluate the average exposure to heavy metals and determine the 95th percentile of exposure from daily intake and for the whole population, of home meal replacement products. The chronic daily intake amount of six heavy metals was 1.60 × 10^−2^ μg/kg/day. Based on total chronic daily intake values, the risk and margin of exposure of each of the heavy metals was 9.13 × 10^7^, demonstrating that intake associated with home meal replacement products is negligible.

## 1. Introduction

In addition to food intake through home cooking, domestic eating patterns have diversified into the consumption of various processed foods, consumption through food service establishments, and consumption through group catering establishments [1]. In particular, in most developed countries, the time to cook at home is decreasing and the frequency of eating out at restaurants or consuming home meal replacements (HMRs) is increasing [2]. HMRs are processed foods that can replace the main course of a meal at home and require minimal preparation prior to consumption [3]. As HMR consumption increases, the exposure of HMRs to hazardous contaminants is also a concern. There is a possibility of exposure to hazardous pollutants in the process of manufacturing, processing, and cooking food ingredients constituting HMRs, and there is also the possibility of exposure to hazardous substances derived from the environment [4]. Although the amount of harmful pollutants generated during, and remaining in food after the manufacturing, processing or cooking of HMR food is very small, the problem of risks associated with these pollutants is attracting much attention from the public as they are consumed throughout life. The media has also raised the issue of food safety regarding various harmful pollutants generated during manufacturing and processing, raising public anxiety about food and increasing consumer distrust [5]. Therefore, research on the product safety of HMR is essential, but not much study is being carried out at present. This study intends to focus specifically on heavy metals that can be introduced into food materials through various routes in HMR products, which are composed of various food groups.

Heavy metals are representative substances that not only exist in nature but are introduced into ecosystems through human activities in their development and normal growth. However, heavy metals such as lead (Pb), cadmium (Cd), arsenic (As), tin (Sn), and mercury (Hg) are highly toxic when consumed in excess and as such are dangerous and not essential for metabolism [6,7]. Pb is used in the manufacture of products such as pesticides, putties, paints and residential installations. It easily enters the body in a number of ways, including through foods such as wine, seafood, fruits, meat, and vegetables, and is extremely harmful to human health [8]. Cd is strongly absorbed by organic substances that make up the soil and is present in the Earth’s crust. This itself poses the greatest risk as these Cd deposits found in organic matter in the soil become part of the food chain for animals and humans. Foods vulnerable to Cd contamination are drinking water, dried seaweed, freshwater fish, shellfish, and mushrooms [9]. As was historically the main cause of serious heavy metal poisoning in humans. The As compounds are found mainly in fish and shellfish [10]. Sn is the main raw material for canning and an ideal coating metal. Hg is a one-of-a-kind type of highly toxic heavy metal and may exist naturally in the form of Hg salt. Hg is discharged into the environment from natural or artificial sources in the form of inorganic Hg compounds, but it can be converted to methylmercury (Me-Hg) in the form of organic substances due to the activity of microorganisms in the environment, such as those found in soil [11]. Me-Hg becomes concentrated in higher organisms through the food chain, so high concentrations can accumulate in fish that predate on others, and then of course can cause harm to humans who consume them. In the general population, the greatest amount of Hg exposure comes from food and the food that contributes most to this intake is fish and fish products [12].

The main sources of heavy metal contamination in food are air, water, and soil [6]. In addition, heavy metals discharged into the environment may be incorporated during the growing or aquaculture of various foods such as fruits, vegetables, grains, livestock, and aquatic products [13]. The increase in wastes generated by various industries, the inflow of wastewater and atmospheric dusty compost, and the excessive use of agricultural chemicals also increase the heavy metal content of agricultural soil, which leads to heavy metal contamination of agricultural products [14]. Recently, as social interest in the safety of agricultural products has increased, domestic and international organizations such as CODEX and EU are lowering the standards for heavy metals in agricultural products and livestock [15]. In particular, since it is often difficult to artificially control the content of heavy metals in food, exposure assessment and safety management are essential. As mentioned earlier, as HMR products become more and more popular, it is necessary to test for the mentioned heavy metal substances. However, there are currently insufficient data on heavy metal risk assessment using HMR products as samples. The aim of this study was to perform chemical analyses for investigation and risk assessment with regard to six heavy metals in HMR, which are increasingly popular and reflect food consumption trends in Korea.

## 2. Materials and Methods

### 2.1. Reagents and Materials

Distilled water was purified through an Fpwps501 Ultrapure Distilled Water Purification System from Human Science (Hanam, Gyeonggi-do, Korea). Hydrogen peroxide (H_2_O_2_, 30%) was purchased from Merck (Darmstadt, Germany). Ultrapure nitric acid (HNO_3_, 70%) was purchased from Duksan (Ansan, Gyeonggi-do, Korea).

Polytetrafluoroethylene membrane filters (0.45 µm) were obtained from Advantec Co., Ltd. (Chiyoda City, Japan). Sodium chloride (NaCl, 25%) was purchased from Samchun (Pyeongtaek, Gyeonggi-do, Korea). Hydrochloric acid (HCl, 98%) was purchased from Duksan (Ansan, Gyeonggi-do, Korea). Toluene (C_6_H_5_CH_3_, 99.5%) was purchased from SK Chemicals (Nam-gu, Ulsan, Korea). L-cysteine (99.0%) was purchased from Samchun (Pyeongtaek, Gyeonggi-do, Korea). The mixed standard solutions for making calibration curves for Pb, Cd, and As were prepared by serial dilution using 7% HNO_3_ in concentrations ranging from 0.25 µg/L to 10.00 µg/L. The mixed standard solutions for making the Sn calibration curve were prepared by serial dilution using 7% HNO_3_ in concentrations ranging from 0.06 mg/L to 1.00 mg/L. Certified Reference Material (CRM) Standard concentrations (1.12 mg/kg of Pb, 0.80 mg/kg of Cd, 13.66 mg/kg of As, 55.43 µg/kg of Hg, and 23.81 µg/kg of Me-Hg) were purchased from the National Institute of Standards and Technology (Gaithersburg, MD, USA). Sn (224.19 mg/kg) was purchased from Korea Research Institute of Standards and Science (Daejeon, Korea).

### 2.2. Sample Preparation for Evaluation of Heavy Metals

From January to September 2020, samples of HMR products were purchased from various convenience and grocery stores located in the Republic of Korea to evaluate the heavy metal content. After removing the non-edible part of each product, the food samples were homogenized with a blender and approximately 10 g stored in zipper bags at −18 °C before digestion and analysis. The samples were sorted into four groups. The first group was Ready to Eat (RTE) products consisting of 100 samples, including ready-to-eat food, fresh convenience food, other processed seafood products, and infants and toddler’s food. The second group was Ready to Heat (RTH) products consisting of 200 samples, including instant cooked foods, sauces, breads, rice cakes, infants and toddler’s food, and other processed seafood products. The third group was Ready to Cook (RTC) products consisting of 140 samples, including meat extract processed products, seasoned meat, infants and toddler’s food, ground processed meat products, and other processed seafood products. The fourth group was Ready to Prepare (RTP) products consisting of 40 samples, including meal kit products.

### 2.3. Sample Digestion and Preprocessing for Pb, Cd, As and Sn Analysis

Fast and convenient sample digestion was conducted by a Terminal 640 microwave-assisted acid digestion system (Milestone, Bergamo, Lombardia, Italy) equipped with a rotor for ten MKM023 HPV-100 TFM vessels. Samples (0.5 g) were placed in a Teflon vessel with 7 mL of ultrapure HNO_3_ (70%) and 0.5 mL of H_2_O_2_ (30%). The vessels were pre-disassembled using a heating block device and distilled water (2 mL) was then added to the pre-disassembled sample and microwave-assisted acid digestion system. Ten vessels were set and operated in the microwave-assisted acid digestion system. The temperature of the microwave was raised to 180 °C for 15 min and maintained for a further 15 min to complete digestion. The vessels were ventilated for 50 min. After cooling at 40 °C for 30 min, ultrapure water was added to the preprocessed solution and volumes diluted to 25 mL. The resulting solutions were filtered through 0.45 µm polytetrafluoroethylene membrane filters for inductively coupled plasma mass spectrometry (ICP-MS) analysis.

### 2.4. Instrument Optimization for ICP-MS Analysis

Heavy metals ICP-MS analysis was performed with a Thermo Fisher Scientific iCAP-RQ equipped with nebulizer, a Teflon spray chamber and nickel (Ni) sampling cone, and platinum skimmer cone (Thermo Fisher Scientific, Waltham, MA, USA). The solutions were pumped by a peristaltic pump from tubes accompanied by an autosampler ASX-560 (ThermoFisher Scientific, Waltham, MA, USA). After the ICP-MS was stabilized for 20–30 min, the working ability was optimized daily with the tuning solution based on torch horizontal and vertical position, extraction lens, CCT (collision cell technology) focus lens and radio frequency power at 1550 W to minimize interference effects and to maximize signal. Highest purity argon and helium gas (99.99%) were used as the carrier gas in auxiliary flow at 0.8 mL/min, and nebulizer flow of 1.0 mL/min and 5.3 mL/min. For quantification, the analyte isotopes ^82^Pb, ^48^Cd, ^33^As and ^50^Sn were used.

### 2.5. Sample Preprocessing for Me-Hg Analysis

Samples (1.0 g) were placed in 50 mL plastic centrifuge tubes, to which was added 10 mL of NaCl (25%), 0.5 mL of HCl (98%) and 15 mL of C_6_H_5_CH_3_ (99.5%). After shaking and extraction for 2 min, centrifugation was performed at 3000 rpm for 20 min. The C_6_H_5_CH_3_ supernatant was aliquoted in a separatory funnel and 5 mL of L-cysteine (99.0%) added with shaking for 10 min. The resulting solutions aliquoted from the L-cysteine (99.0%) layer were placed in quartz sample boats for direct Hg analyzer (DMA) analysis.

### 2.6. Instrument Optimization for DMA Analysis

Heavy metal DMA analysis was performed with a DMA (DMA-80 evo, Milestone, Waltham, Sorisole, Italy). DMA was based on the electrothermal atomization of Hg due to sample drying and subsequent pyrolysis. A number of quartz sample boats (Milestone, DMA 8347) were set up for analytical method development and validation, and a Ni sample boat was set up for robustness testing. An internal thermocouple sensor modulated decomposition/drying temperature. For Hg vapor wiring a gold amalgamator system was set up with an infrared temperature sensor (Milestone, DMA 8134). A gold amalgamator was set up to selectively capture and pre-concentrate Hg in the decomposition products stream. A catalytic system was set up for Hg reduction (Milestone, DMA 8333). Thermally desorbed elemental Hg was quantified using an optical cell for detection through atomic absorption spectroscopy at 253.65 nm. The detection system included a silicon UV diode detector, low-pressure Hg vapor lamp, Hg-specific lamp and dual-cell spectrophotometer with a wide working range. Peak heights were expressed for signal evaluation. For quantification, the analyte isotope ^80^Hg was used.

### 2.7. Method Validation for Quality Assurance of Analysis

The ICP-MS and DMA analytical methods were validated for recovery (%), precision (%), accuracy (%), limits of quantification (LOQ), limits of detection (LOD), linearity (*R*^2^) and measurement of uncertainty (MU) on HMR product samples. Calibration curves were obtained by the analysis of Pb, Cd, As and Sn standard mixtures at 0.25, 1.0, and 10.0 mg/kg. Calibration curves were obtained by the analysis of Hg, Me-Hg standard mixture in absolute amounts at 5.0, 20.0, and 100.0 μg/kg. These validation parameters followed the protocol of CODEX guidelines except for MU [15]. The MU followed the protocol of EURACHEM guidelines [16]. To obtain accuracy estimations, five different concentrations (used for the calibration curve) were tested in three repetitions both intra- and inter-day. Among them, only values obtained at concentrations higher than LOQ were used for each element. Accuracy was calculated from the function shown in Equation (1).
Accuracy (%) = (C_mean_ − C_blk_)/C_spiked_ × 100 (%)(1)
where C_mean_, C_blk_, and C_spiked_ are the average concentration of standard, blank, and spiked, respectively. The precision was obtained using a coefficient of variation (CV). To estimate precision, standard solutions were spiked into samples. Three different samples were repeated with three different concentrations, and the highest values were adopted. The CV was estimated from the function of Equation (2).
CV (%) = C_sd_/C_mean_ × 100 (%)(2)
where C_sd_ is the standard deviation of the concentration of standard. The standard uncertainty of recovery μ(Rec) was determined by using Equation (3).
(3)$$\mu (\mathrm{Rec})=\mathrm{Recovery}\;\times \;\sqrt{\frac{{\left(C_{\mathrm{sd}}\right)}^{2}}{n\times {\left(C_{\mathrm{mean}}\right)}^{2}}}$$

The relative standard uncertainty of recovery (RSU) and the combined standard uncertainty (CSU) were estimated from the function in Equations (4) and (5).
(4)$$\mathrm{RSU}=\frac{\mu \left(\mathrm{Rec}\right)}{\mathrm{Recovery}}\times 100\%$$
(5)$$\mathrm{CSU}=\sqrt{{\left(\mathrm{Precision}\right)}^{2}+{\left(\mathrm{bias}\right)}^{2}}$$

The expanded uncertainty of recovery (EU) was calculated by using a coverage factor (k) of 2 at a 95% confidence level.

### 2.8. Exposure Assessment

This part of the study was conducted by analyzing the exposure dose of the living population from the research data obtained. Dividing ages into bins of 1–2, 3–6, 7–12, 13–19, 20–64, >65 and all age years, heavy metal daily exposure from food was carried out. For this purpose, the weight by age group to be evaluated and the latest raw data from the National Health and Nutrition Survey provided by the Korea Centers for Disease Control and Prevention were used.
(6)$$95\mathrm{th}\;\mathrm{percentile}\;\mathrm{food}\;\mathrm{intake}\;\mathrm{data}\;(g/\mathrm{day})=\mathrm{percentile}\;({\int_{1}}^{j}\;\sum_{i=1}^{n}\left[IRi\right],\;0.95)$$
where IRi is the amount of food for each item of food i for daily intake acquired from the National Health and Nutrition Survey (g/day); *n* is the number of individual items and j is the number of respondents by age group.
(7)$$95\mathrm{th}\;\mathrm{percentile}\;\mathrm{CDI}\;(\mu g/\mathrm{kg}/\mathrm{day})=\mathrm{percentile}\;({\int_{1}}^{j}\;\sum_{i=1}^{n}\frac{Ci\times IRi}{\mathrm{BW}},\;0.95)$$
where CDI is the chronic daily intake; *Ci* is the concentrations of the six heavy metals in the food *i* (μg/kg); IRi is the amount of food for each item of food i for daily intake, acquired from the National Health and Nutrition Survey (g/day); BW is the average body weight by age (58.50 kg); j is the number of respondents by age group; and n is the number of individual items (Center for Disease Control, Cheongju, Korea).

### 2.9. Risk Characterization

The risk of food heavy metal CDI was evaluated according to the following equation by comparison with the human exposure safety standard of the target substance.
(8)$$\mathrm{Risk}=\frac{\mathrm{CDI}}{PTWI}$$
where *PTWI* is the human exposure safety standard (μg/kg BW/day).

In the case of Pb among the target substances, the margin of exposure (MOE) was calculated with the CDI (μg/kg BW/day) and the benchmark dose lower confidence limit (BMDL) (μg/kg BW/day). The BMDL was the lower limit of the 99% confidence interval of the BMDL at the dose resulting in a change in biological effect, or a predetermined measure for toxicity resulting in an adverse effect of 10% or 1% compared with the control group.
(9)$$\mathrm{MOE}=\frac{\mathrm{BMDL}01}{\mathrm{CDI}}$$

### 2.10. Statistical Analysis

All analyses were executed in triplicate, and the data are expressed as mean ± standard deviation (SD).

## 3. Results

### 3.1. Method Validation for Heavy Metals Analysis

Calibration curves were determined for the six heavy metals at five different concentrations (0.25, 0.5, 1.0, 5.0, and 10.0 mg/kg for Pg, Cd, As, and Sn and 5.0, 10,0, 20.0, 50.0, and 100.0 μg/kg for Hg and Me-Hg). Linearity is then expressed in the slope of the plot of area value of the chromatogram in proportion to the concentration and is expressed as the coefficient of determination (*R*^2^). Linearity was good in the concentration range tested, with *R*^2^ values higher than 0.99. For the non-fatty solid, fatty solid, non-fatty liquid, and fatty liquid matrix, the LOQ ranged from 0.05 to 49 μg/kg and the LOD ranged from 0.02 to 16 μg/kg. The LOQ, LOD, and linearity of the six heavy metals are presented in Table 1.

The recovery values of the heavy metals (Appendix A) were expressed by using the accuracy values. The recovery values of heavy metals in the non-fatty solid, fatty solid, non-fatty liquid, and fatty liquid matrices ranged from 80.65% to 119.87%. Accuracy and precision were measured at different concentrations (0.25, 1.0, and 10.0 mg/kg for Pg, Cd, As, and Sn, and 5.0, 20.0, and 100.0 μg/kg for Hg and Me-Hg). The accuracy and precision of heavy metal analyses in the non-fatty solid, fatty solid, non-fatty liquid, and fatty liquid matrices were in the ranges of 80.49–119.49% and 0.26–6.00% (intra-day), and 80.33–119.61% and 0.79–14.93% (inter-day), respectively. Corresponding data for the six heavy metals are presented in Table 2 and Table 3.

### 3.2. Estimation of Measurement Uncertainty

The recovery data obtained were further used in the calculation of the uncertainty of recovery. The results of μ(Rec), RSU, CSU, and EU are summarized in Appendix A. The μ(Rec) ranged from 0.082% to 0.321% and the RSU ranged from 0.084% to 0.320%. Expanded uncertainty of recovery was calculated by using a coverage factor of 2 at a 95% confidence level.

### 3.3. External Quality Assurance

The accuracy of the analysis was verified by the FAPAS proficiency test October-November 2020 ‘Metallic Contaminants in Soya Flour from Fera Science Ltd.’ (Sand Hutton, York, UK). The comparisons of the certified and measured values of six heavy metals in the CRM (accuracy) are shown in Table 4.

### 3.4. Comparisons of Heavy Metal Content in HMR Products

The concentrations of heavy metals were determined for 480 samples of HMR products (Table 5).

The mean concentration level of heavy metals was 8.87 μg/kg. Among HMR samples, the As value of the non-fatty solid phase had the highest concentration of any heavy metal (0.154 mg/kg). Considering the non-fatty solid phase samples, the mean concentrations of heavy metals were 12.26 μg/kg. For fatty solid-phase samples, the mean concentrations of heavy metals were 4.29 μg/kg, while for the non-fatty liquid-phase samples, the mean concentrations of heavy metals were 11.17 μg/kg. The fatty liquid-phase samples had mean concentrations of heavy metals of 7.75 μg/kg.

### 3.5. Exposure Assessment

The total average food intake and 95th percentile food intake data (g/day) are detailed in Appendix A. These data were calculated based on the amount of food for each item, the daily intake acquired from the National Health and Nutrition Survey (g/day), the number of individual items, and the number of respondents by age group in Korea. In HMR products, values were 0.088 g/day for non-fatty solid phase, 0.027 g/day for fatty solid phase, 0.020 g/day for non-fatty liquid phase, and 0.005 g/day for fatty liquid phase as overall averages. Following estimation of the 95th percentile food intake data (g/day), the exposure assessment of all heavy metals was calculated based on the heavy metal concentrations and 95th percentile food intake data (g/day). Age groups were classified as 1–2, 3–6, 7–12, 13–19, 20–64, and >65 years and CDI was evaluated by BW and exposure duration. The results showe that for Korean people of all age groups, consuming HMR products led to average heavy metal totals from food, of 0.023 g/day and 0.047 g/day. Correspondingly, in the case of HMR products, the average CDI of all heavy metals was 1.06 × 10^−2^ μg/kg/day. Furthermore, with high consumption of HMR products (95th percentile), the average CDI was 2.15 × 10^−2^ μg/kg/day. The CDI of heavy metals in HMR products is presented in Table 6 and the average body weights and exposure periods for calculating chronic daily intake in HMR products are detailed in Appendix A.

### 3.6. Risk Characterization

From the CDI results of the analyzed HMR products, the risk of all heavy metals was characterized by calculating MOE values for Koreans, as shown in Appendix A. The MOE values were 1.48 × 10^8^ for the total population and 3.46 × 10^7^ for the 95th percentile group.

## 4. Discussion

By using ICP-MS with DMA, it was possible to accurately and precisely measure the heavy metals contaminated in HMR, and excellent selectivity, sensitivity, and resolution of analysis methods were demonstrated in this study. As a result, a new method for ICP-MS and DMA analysis of heavy metals was developed by ensuring that the validation, investigation, and risk assessment results correlated well.

The high linearity (*R*^2^ > 0.99) of our method validation is similar to that reported for other spiked food matrices, including cocoa beans, cocoa products and chocolate [17]. The LOQ and LOD results of our validation tests are lower than those of heavy metals in cocoa beans, cocoa products, and chocolate, which ranged from 0.2 to 69.3 μg/kg for LOQ and 0.1–20.8 μg/kg for LOD. It was confirmed that the LOQ and LOD were relatively high in the fatty matrix and low in the non-fatty matrix. The reason for this observation is that the concentrations of harmful substances formed from the fatty matrix increase during cooking, manufacturing, and processing [18]. Recovery results were analogous between the fatty and non-fatty matrix. The recovery value using the efficiency of extraction time through vortex mixing is in the range of 80–120%. These recovery data demonstrate that vortex mixing was more efficient than the ultrasonic bath experiment for extraction, regardless of the matrix type [19]. Accuracy and precision were also analogous between the fatty and non-fatty matrices. Through the results of Table 2 and Table 3, it was identified that the accuracy and the precision demonstrated by our analysis method meet the necessary criteria. In addition, it was confirmed that for the intra-day analysis, carried out quickly three times a day, and for the inter-day analysis, performed at the same time on three days, accuracy was in the range of 80–120% and precision (%CV) was <15%. These data reflect the efficiency of the method for non-fatty solid, fatty solid, non-fatty liquid, and fatty liquid matrices. According to the CODEX and EURACHEM guidelines, the analysis of all heavy metals exhibited values that were validated for these substances [10,15,16]. Dico et al. [20] performed method validation of ICP-MS detection for the determination of selected heavy metal fractions in some infant liquid milk formulas and obtained LOD of 0.001–0.50 mg/kg, LOQ of 0.002–1.00 mg/kg, and recovery of 96.00–105.00%, data comparable to the current method. Habte et al. [21] developed and validated a high-resolution DMA method for the analysis of heavy metals in HMR products. They reported mean values for LOD and LOQ of 0.004 and 0.008 ng/g, respectively. Recoveries of heavy metals were in the range of 88.60–105.30% for Ethiopian coffee samples. They mentioned that the heavy metal contents of HMR products are affected by various factors, including the temperature of combustion, the temperature of oxidation and relative humidity. Table 4 indicates that the methods developed in this study pass the proficiency test conducted by FAPAS. To obtain information on the degree of accuracy, the CRM test was replicated three times. For the six heavy metals, it ranged from 84.76% to 102.71%, and the relative standard deviation (RSD) of recovery ranged from 1.86% to 4.96%. The accuracy and RSD, therefore, satisfy the method validation criteria and indicate the good accuracy and RSD of the CRM proficiency test.

In Korea, there are no set standards for the six heavy metals related to HMR products. However, all of the samples were identified as having safe levels of six heavy metals in comparison with processed food outliers. The causes of the formation and contamination of heavy metal HMR products are mainly contaminated soil where the food ingredients are grown, and the chemical pesticides and fertilizers used in food cultivation. Heavy metals are naturally occurring compounds which have high density compared with other metals, are non-biodegradable and are five times denser than water [22]. Besides formation and contamination issues, food containing heavy metals have the potential to be toxic to humans when ingested in small amounts [23]. These heavy metals cause toxicity by creating complexes with nitrogen-containing cells, oxygen, and sulfur compounds [24]. Moreover, toxic heavy metals in these HMR products may be consumed by humans and cause cancer [25]. The confirmed concentrations of heavy metals in the HMR products we tested were relatively low and similar to cassava, cocoyam, plantain, and yam, relative to unprocessed foods or foods that have been cooked or laboratory processed [26,27]. Processed foods have high heavy metal levels depending on several factors. For instance, the content of heavy metals in processed cocoa bean products is promoted by protecting the core from chemical, external physical threats and a defense mechanism created by peeling during cocoa bean processing [28]. Overall, the heavy metal contents found in processed foods are higher than those contents found in the HMR products in our study. In light of the published results on heavy metals in processed foods containing fatty ingredients, our results suggest that HMR products contain lower amounts of heavy metals. This is similar to the results of the study of Shariatifar et al. [29]. They found that the content of heavy metals in refined salt was lower than the maximum amount set by CODEX. In addition, there are previous studies that measured the concentration of heavy metals in baby food, milk powder, and edible mushrooms [30,31]. Additionally, all of them confirmed the presence of heavy metals at safe levels in each food through risk assessment.

The MOE value suggests that the contamination levels of heavy metals in the HMR category do not pose a public health risk. According to the MOE and CDI values, MOE decreases in proportion to CDI in the case of Pb. In contrast to these values, MOE increases in proportion to CDI for Cd, As, Sn, Hg, and Me-Hg. The MOE values of heavy metals in HMR products were similar to the MOE values of heavy metals in different contaminated food crops (maize grains, rice grains, wheat grains and mustard seeds), which ranged from 2.62 × 10^5^ to 3.24 × 10^8^ [25]. Contaminated food crop products had similar results to data obtained from this study at around 4.46 × 10^−5^ to 5.68 × 10^−2^. This was because HMR products were less exposed to heavy metals than other processed products.

## 5. Conclusions

In this study, the quantification of six heavy metals (Pb, Cd, As, Sn, Hg, and Me-Hg) present in HMR products via a microwave acid digestion system and analyzed by ICP-MS and DMA was successfully validated. Through the study, it was proven that this analysis method is a convenient, fast, and reliable process for measuring trace elements and heavy metals in HMRs. Additionally, it has demonstrated acceptable linearity, LOD, LOQ, accuracy, precision, and recovery in terms of both the validation criteria and the requirements set by the CODEX and EURACHEM guidelines. Compared with other papers, the validation results of this study were acceptable. Overall, the results showed very low levels in the risk assessment and concentrations of heavy metal contaminants in HMR products, so it was concluded that heavy metal levels in HMRs are within safe limits. In order to provide consumers with HMR products that are safe from heavy metals, safety management through continuous exposure assessment and analysis is required.

## Figures and Tables

**Table 1 foods-11-00504-t001:** The LOD, LOQ, linearity equations, and *R*^2^ of the six heavy metals.

Matrix Type	Heavy Metal Element	LOD ^a^ (µg/kg)	LOQ ^b^ (µg/kg)	Linearity Equation ^c^	*R* ^2^
Non-fatty solid phase	Pb	0.09	0.29	y = 107,991x + 44,824	0.9997
	Cd	0.03	0.08	y = 44,515x − 1453.8	0.9984
	As	0.02	0.05	y = 2415.7x + 107.13	0.9998
	Sn	16.00	49.00	y = 10,000,000x − 206,653	0.9989
	Hg	2.79	8.46	y = 0.0172x + 0.0425	0.9977
	Me-Hg	1.34	4.06	y = 0.0066x + 0.01	0.9989
Fatty solid phase	Pb	0.09	0.29	y = 108,635x + 43,525	0.9998
	Cd	0.03	0.08	y = 44,439x − 1505.8	0.9997
	As	0.02	0.05	y = 11,253x − 344.17	0.9919
	Sn	4.00	11.00	y = 10,000,000x − 107,914	0.9978
	Hg	0.63	1.92	y = 0.0171x + 0.0441	0.9979
	Me-Hg	0.67	2.04	y = 0.007x + 0.0219	0.9989
Non-fatty liquid phase	Pb	0.09	0.29	y = 106,669x + 46,240	0.9995
	Cd	0.03	0.08	y = 43,756x − 1309.4	0.9902
	As	0.02	0.05	y = 10,975x − 292.49	0.9968
	Sn	11.00	34.00	y = 10,000,000x + 100,280	0.9985
	Hg	1.56	4.72	y = 0.0173x + 0.0395	0.9994
	Me-Hg	0.23	0.69	y = 0.0072x + 0.0221	0.9961
Fatty liquid phase	Pb	0.09	0.29	y = 108,669x + 44,707	0.9997
	Cd	0.03	0.08	y = 45,349x − 1546.1	0.9943
	As	0.02	0.05	y = 11,325x − 331.02	0.9952
	Sn	1.00	30.00	y = 10,000,000x − 313,065	0.9966
	Hg	2.81	8.51	y = 0.0173x + 0.0162	0.9939
	Me-Hg	1.75	5.30	y = 0.0072x + 0.0252	0.9988

^a^ set up in a signal-to-noise ratio (S/N) = 3.3; ^b^ set up in a signal-to-noise ratio (S/N) = 10; ^c^ numbers express mean values (*n* = 3).

**Table 2 foods-11-00504-t002:** Accuracy analysis of the six heavy metals.

Matrix Type	Heavy Metal Element	Accuracy (%)					
		Intraday (*n* = 3)			Interday (*n* = 3)		
Non-fatty solid phase	Pb (mg/kg)	103.07	80.49	101.50	93.18	101.80	103.22
	Cd (mg/kg)	89.66	101.04	100.15	96.81	117.80	100.28
	As (mg/kg)	107.11	109.47	97.26	106.90	107.36	99.69
	Sn (mg/kg)	82.00	81.00	93.80	85.33	80.33	90.20
	Hg (µg/kg)	99.02	101.57	99.27	98.68	98.09	95.34
	Me-Hg (µg/kg)	102.51	95.16	102.05	103.34	105.14	111.43
Fatty solid phase	Pb (mg/kg)	103.07	80.49	101.50	93.18	101.80	103.22
	Cd (mg/kg)	109.66	81.04	100.15	96.81	107.80	100.28
	As (mg/kg)	107.11	109.47	97.26	106.90	107.36	99.69
	Sn (mg/kg)	86.00	87.00	90.40	92.00	88.67	90.87
	Hg (µg/kg)	92.50	93.88	96.94	96.21	94.49	97.48
	Me-Hg (µg/kg)	115.62	114.32	107.93	108.83	113.15	106.66
Non-fatty liquid phase	Pb (mg/kg)	103.07	80.49	101.50	93.18	101.80	103.22
	Cd (mg/kg)	109.66	101.04	100.15	96.81	107.80	100.28
	As (mg/kg)	107.11	109.47	97.26	106.90	107.36	99.69
	Sn (mg/kg)	96.00	109.00	107.00	86.67	95.00	98.93
	Hg (µg/kg)	97.77	93.91	93.96	98.49	94.78	95.48
	Me-Hg (µg/kg)	105.47	114.27	114.17	103.79	112.58	111.16
Fatty liquid phase	Pb (mg/kg)	103.07	80.49	101.50	93.18	101.80	103.22
	Cd (mg/kg)	109.66	101.04	100.15	96.81	107.80	100.28
	As (mg/kg)	107.11	109.47	97.26	106.90	107.36	99.69
	Sn (mg/kg)	82.00	81.00	88.00	83.33	85.67	100.60
	Hg (µg/kg)	92.19	90.89	92.96	89.88	93.49	94.64
	Me-Hg (µg/kg)	116.13	119.49	116.02	119.61	115.04	112.87

**Table 3 foods-11-00504-t003:** Precision of analysis (% CV) of the six heavy metals.

Matrix Type	Heavy Metal Element	Precision (%)					
		Intraday (*n* = 3)			Interday (*n* = 3)		
Non-fatty solid phase	Pb (mg/kg)	4.29	3.68	2.48	5.06	1.71	4.80
	Cd (mg/kg)	2.64	3.34	1.22	2.45	0.94	1.64
	As (mg/kg)	5.16	3.09	1.40	2.66	4.92	2.85
	Sn (mg/kg)	5.24	4.48	2.54	8.07	4.29	11.97
	Hg (µg/kg)	2.34	1.05	0.39	1.95	2.79	3.79
	Me-Hg (µg/kg)	6.00	3.25	1.10	4.94	7.53	9.30
Fatty solid phase	Pb (mg/kg)	4.29	3.68	2.48	5.06	1.71	4.80
	Cd (mg/kg)	2.64	3.34	1.22	2.45	0.94	1.64
	As (mg/kg)	5.16	3.09	1.40	2.66	4.92	2.85
	Sn (mg/kg)	3.07	0.93	1.29	11.56	13.85	7.45
	Hg (µg/kg)	0.53	0.26	0.49	4.47	3.62	3.61
	Me-Hg (µg/kg)	1.11	0.61	1.26	10.42	8.63	9.52
Non-fatty liquid phase	Pb (mg/kg)	4.29	3.68	2.48	5.06	1.71	4.80
	Cd (mg/kg)	2.64	3.34	1.22	2.45	0.94	1.64
	As (mg/kg)	5.16	3.09	1.40	2.66	4.92	2.85
	Sn (mg/kg)	2.11	3.54	1.72	11.70	14.33	14.93
	Hg (µg/kg)	0.36	0.40	0.39	2.58	1.82	2.04
	Me-Hg (µg/kg)	0.88	0.94	0.92	6.49	4.38	5.04
Fatty liquid phase	Pb (mg/kg)	4.29	3.68	2.48	5.06	1.71	4.80
	Cd (mg/kg)	2.64	3.34	1.22	2.45	0.94	1.64
	As (mg/kg)	5.16	3.09	1.40	2.66	4.92	2.85
	Sn (mg/kg)	5.29	3.00	1.95	13.18	11.16	10.38
	Hg (µg/kg)	2.70	1.96	1.05	0.79	2.27	2.57
	Me-Hg (µg/kg)	5.57	4.18	2.39	1.53	5.26	6.17

**Table 4 foods-11-00504-t004:** Comparisons of the certified and measured values of six heavy metals in the CRM (accuracy).

Heavy Metal Element	CRM	Certified Value	Measured Value	Accuracy (%)
Pb (mg/kg)	NIST SRM 2976 (Mussel tissue)	1.19	1.12	94.12
Cd (mg/kg)		0.82	0.80	97.56
As (mg/kg)		13.30	13.66	102.71
Sn (mg/kg)	KRISS CRM 108-05-005 (Tomato paste)	221.20	224.19	101.35
Hg (µg/kg)	NIST CRM 2976 (Mussel tissue)	61.00	55.43	90.87
Me-Hg (µg/kg)		28.09	23.81	84.76

**Table 5 foods-11-00504-t005:** Heavy metal concentrations in non-fatty solid phase, fatty solid phase, non-fatty liquid phase and fatty liquid phases of HMR.

Matrix Type	Heavy Metal Element	Mean Value	Min Value	Max Value	Median Value
Non-fatty solid phase	Pb (mg/kg)	0.013	0.002	0.089	0.015
	Cd (mg/kg)	0.013	ND ^a^	0.110	0.055
	As (mg/kg)	0.039	ND	0.154	0.077
	Sn (mg/kg)	ND	ND	ND	ND
	Hg (µg/kg)	5.533	ND	24.667	12.333
	Me-Hg (µg/kg)	3.000	ND	48.000	24.000
Fatty solid phase	Pb (mg/kg)	0.013	0.003	0.087	0.045
	Cd (mg/kg)	0.003	ND	0.017	0.009
	As (mg/kg)	0.009	ND	0.051	0.025
	Sn (mg/kg)	ND	ND	ND	ND
	Hg (µg/kg)	0.760	ND	14.250	7.125
	Me-Hg (µg/kg)	ND	ND	ND	ND
Non-fatty liquid phase	Pb (mg/kg)	0.010	0.001	0.044	0.023
	Cd (mg/kg)	0.007	ND	0.117	0.059
	As (mg/kg)	0.048	ND	0.108	0.054
	Sn (mg/kg)	ND	ND	ND	ND
	Hg (µg/kg)	2.000	ND	50.000	25.000
	Me-Hg (µg/kg)	ND	ND	ND	ND
Fatty liquid phase	Pb (mg/kg)	0.012	0.005	0.062	0.033
	Cd (mg/kg)	0.004	ND	0.017	0.008
	As (mg/kg)	0.030	ND	0.055	0.028
	Sn (mg/kg)	ND	ND	ND	ND
	Hg (µg/kg)	0.500	ND	9.000	4.500
	Me-Hg (µg/kg)	ND	ND	ND	ND

^a^ ND = not detected, the lower limit of detection.

**Table 6 foods-11-00504-t006:** CDI of the six heavy metals in non-fatty solid phase, fatty solid phase, non-fatty liquid phase and fatty liquid phases of HMR.

Matrix Type	Age (Year)	Average Dietary Exposure (μg/kg/day)	95th Percentile Dietary Exposure (μg/kg/day)
Non-fatty solid phase	Above 65	1.04 × 10^−2^	3.41 × 10^−2^
	20–64	9.32 × 10^−3^	3.07 × 10^−2^
	13–19	1.02 × 10^−2^	3.34 × 10^−2^
	7–12	1.60 × 10^−2^	5.28 × 10^−2^
	3–6	3.08 × 10^−2^	1.01 × 10^−1^
	1–2	4.80 × 10^−2^	1.58 × 10^−1^
Fatty solid phase	Above 65	2.05 × 10^−3^	3.94 × 10^−3^
	20–64	1.84 × 10^−3^	3.54 × 10^−3^
	13–19	2.01 × 10^−3^	3.86 × 10^−3^
	7–12	3.17 × 10^−3^	6.10 × 10^−3^
	3–6	6.08 × 10^−3^	1.17 × 10^−2^
	1–2	9.48 × 10^−3^	1.82 × 10^−2^
Non-fatty liquid phase	Above 65	8.53 × 10^−3^	2.98 × 10^−3^
	20–64	7.67 × 10^−3^	2.68 × 10^−3^
	13–19	8.36 × 10^−3^	2.92 × 10^−3^
	7–12	1.32 × 10^−2^	4.61 × 10^−3^
	3–6	2.53 × 10^−2^	8.85 × 10^−3^
	1–2	3.95 × 10^−2^	1.38 × 10^−2^
Fatty liquid phase	Above 65	2.01 × 10^−4^	1.81 × 10^−3^
	20–64	1.81 × 10^−4^	1.62 × 10^−3^
	13–19	1.97 × 10^−4^	1.77 × 10^−3^
	7–12	3.11 × 10^−4^	2.80 × 10^−3^
	3–6	5.96 × 10^−4^	5.37 × 10^−3^
	1–2	9.30 × 10^−4^	8.37 × 10^−3^

## Data Availability

Not applicable.

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
