# Peer review of "Risk Assessment and Determination of Heavy Metals in Home Meal Replacement Products by Using Inductively Coupled Plasma Mass Spectrometry and Direct Mercury Analyzer"

_foods, 2022, doi:10.3390/foods11040504_

Round 1
Reviewer 1 Report
In my opinion, the English language expressions in the manuscript require copyediting. The writing of this manuscript is not easy for readers to understand. Some sentences fail to express the original meaning. I suggest using a professional copyeditor or a native English speaker to copy edit this manuscript.
The novelty of the work should be established.
Introduction, poorly written
Please see the following articles
Probabilistic Health Risk Assessment of Trace Elements in Baby Food and Milk Powder Using ICP-OES Method
The Concentration and probabilistic health risk of potentially toxic elements (PTEs) in Edible Mushrooms (Wild and Cultivated) samples collected from different cities of Iran
Assessment of Rice Marketed in Iran with Emphasis on Toxic and Essential Elements; Effect of Different Cooking Methods.
Assessment of heavy metal content in refined and unrefined salts obtained from Urmia, Iran
The "discussion" section is particularly weak in terms of "literature review".
The conclusion should be concise and to the points indicating the application of the work
Throughout the text, some words must match the format of the journal.
In general, the text has many errors that must be carefully corrected by the author.
Author Response
The whole manuscript was edited by a professional English editor. (www.harrisco.net)). And the manuscript has been revised in its entirety.
We look forward to your positive response.
Thank you.

Reviewer 2 Report
The manuscript is presented a study of determination, validation of heavy metals in home meal replacement. The authors also conducted a risk assessment study for the consumption of these heavy metals in those meals. A major revision is needed.
The points that author should consider are presented below:
Major revision:
- Terminology: The authors refer to EURACHEM guideline for the validation. In this guideline the term “accuracy” is sum of precision (RSD) and trueness (recovery). In manuscript the terms are different, accuracy (recovery) and precision (RSD). This maybe confused readers. The authors should check if it is better the terminology to be in line with the guideline that they refer.
- Uncertainty Concept: The main uncertainty parameters, that they contribute significantly to standard uncertainty of the final result, are uncertainty of random errors (RSD) and bias error (μ(REC)). In the manuscript, the calculation of standard uncertainty (RSU) included only the bias parameters. So authors should re-evaluate the uncertainty including the random error uncertainty and combined all parameters to calculate the standard uncertainty (RSU). Please check the EURACHEM guideline for the uncertainty (Quantifying Uncertainty in Analytical Measurement, 3rd Edition) for further information.
- Table S1: Author refer at footnote that the results are presented as mean +/- standard deviation but they show only mean value. Please revise the Table S1
- Table S1: The different fortification levels include 0 mg/kg level, how the author evaluate the recovery in zero fortification level without any spiking concentration. Please check if there is any error in typing.
Minor Revision
- Significant Figures. The authors should revise the significant figures that the results are presented in all manuscript. The general approach is that all results should have 3 significant figures and limit of detection and quantification 2 significant figure.
- There is a contradiction between the equations in line 185 and 264, for the calculation of Accuracy that it will be confuse readers. Moreover, it is not clear the symbols of equation 1,2. please explain where the symbols refer to because there is a confusion with the equation in lines 264 and 266.
Author Response
- Terminology: The authors refer to EURACHEM guideline for the validation. In this guideline the term “accuracy” is sum of precision (RSD) and trueness (recovery). In manuscript the terms are different, accuracy (recovery) and precision (RSD). This maybe confused readers. The authors should check if it is better the terminology to be in line with the guideline that they refer.
(Answer) The validation parameters in our study followed the protocol of CODEX guidelines except for measurement of uncertainty, and the measurement of uncertainty followed the protocol of EURACHEM guidelines. So, the validation method has been revised (lines 176-178, 344, and 403).
- Uncertainty Concept: The main uncertainty parameters, that they contribute significantly to standard uncertainty of the final result, are uncertainty of random errors (RSD) and bias error (μ(REC)). In the manuscript, the calculation of standard uncertainty (RSU) included only the bias parameters. So authors should re-evaluate the uncertainty including the random error uncertainty and combined all parameters to calculate the standard uncertainty (RSU). Please check the EURACHEM guideline for the uncertainty (Quantifying Uncertainty in Analytical Measurement, 3rd Edition) for further information.
(Answer) We acknowledge that there is no description of "combined standard uncertainty" and "expanded uncertainty" in the manuscript. So, referring to EURACHEM guideline, the data about "combined standard uncertainty" and "expanded uncertainty" have been added in the manuscript (Table S2; lines 198-206, 272).
- Table S1: Author refer at footnote that the results are presented as mean +/- standard deviation but they show only mean value. Please revise the Table S1
(Answer) As the reviewer pointed out, Table S1 has been reviewed and revised in its entirety.
- Table S1: The different fortification levels include 0 mg/kg level, how the author evaluate the recovery in zero fortification level without any spiking concentration. Please check if there is any error in typing.
(Answer) As the reviewer pointed out, Table S1 has been reviewed and revised in its entirety.
Minor Revision
- Significant Figures. The authors should revise the significant figures that the results are presented in all manuscript. The general approach is that all results should have 3 significant figures and limit of detection and quantification 2 significant figure.
(Answer) As the reviewer pointed out, data in the manuscript has been reviewed and revised in its entirety (Table 1; Table S3; lines 300-302).
- There is a contradiction between the equations in line 185 and 264, for the calculation of Accuracy that it will be confuse readers. Moreover, it is not clear the symbols of equation 1,2. please explain where the symbols refer to because there is a confusion with the equation in lines 264 and 266.
(Answer) The lines 264 and 266 have been deleted and explanation of the symbols of equation 1 and 2 have been added in the manuscript (lines 185, 186, and 193).
The manuscript has been resubmitted to your journal. We look forward to your positive response.
Thank you.

Round 2
Reviewer 1 Report
Accept in present form
Reviewer 2 Report
No further comments.